# Effects of Thermal Stress on the Gut Microbiome of Juvenile Milkfish (*Chanos chanos*)

**DOI:** 10.3390/microorganisms9010005

**Published:** 2020-12-22

**Authors:** Christiane Hassenrück, Hannes Reinwald, Andreas Kunzmann, Inken Tiedemann, Astrid Gärdes

**Affiliations:** 1Leibniz Centre for Tropical Marine Research (ZMT), Fahrenheitstraße 6, 28359 Bremen, Germany; andreas.kunzmann@leibniz-zmt.de (A.K.); inken.hanke@leibniz-zmt.de (I.T.); 2MARUM—Center for Marine Environmental Sciences, University of Bremen, Leobener Straße 8, 28359 Bremen, Germany; 3Max Planck Institute for Marine Microbiology, Celsiusstraße 1, 28359 Bremen, Germany; hannes.reinwald@ime.fraunhofer.de; 4Fraunhofer Institute for Molecular Biology and Applied Ecology IME, Auf dem Aberg 1, 57392 Schmallenberg, Germany; 5Alfred-Wegener-Institut, Helmholtz-Zentrum für Polar- und Meeresforschung (AWI), Am Handelshafen 12, 27570 Bremerhaven, Germany; 6Hochschule Bremerhaven, An der Karlstadt 8, 27568 Bremerhaven, Germany

**Keywords:** aquaculture, intestinal microbial communities, temperature stress, energy metabolism, 16S rRNA gene sequencing

## Abstract

Milkfish, an important aquaculture species in Asian countries, are traditionally cultured in outdoor-based systems. There, they experience potentially stressful fluctuations in environmental conditions, such as temperature, eliciting changes in fish physiology. While the importance of the gut microbiome for the welfare and performance of fish has been recognized, little is known about the effects of thermal stress on the gut microbiome of milkfish and its interactions with the host’s metabolism. We investigated the gut microbiome of juvenile milkfish in a thermal stress experiment, comparing control (26 °C) and elevated temperature (33 °C) treatments over three weeks, analyzing physiological biomarkers, gut microbiome composition, and tank water microbial communities using 16S amplicon sequencing. The gut microbiome was distinct from the tank water and dominated by *Cetobacterium*, *Enterovibrio*, and *Vibrio*. We observed a parallel succession in both temperature treatments, with microbial communities at 33 °C differing more strongly from the control after the initial temperature increase and becoming more similar towards the end of the experiment. As proxy for the fish’s energy status, HSI (hepatosomatic index) was correlated with gut microbiome composition. Our study showed that thermal stress induced changes in the milkfish gut microbiome, which may contribute to the host’s habituation to elevated temperatures over time.

## 1. Introduction

Milkfish (*Chanos chanos*) is a euryhaline, mainly herbivorous teleost of the Chanidae family and an important aquaculture species for tropical countries. With an annual global production of more than 1 Mt in the last years [1] and several Southeast Asian countries as main producers, milkfish plays a key role in providing affordable animal-based protein for the local population. Milkfish are traditionally cultured in outdoor-based systems in hypersaline lagoons, brackish-water ponds or marine systems, where they experience daily and/or seasonal variation in ambient conditions [2,3]. For instance, water temperature can reach limits at 15 and 40 °C during cold snaps or heat waves [2]. Being poikilothermic ectotherms, fluctuations in water temperature have a direct effect on many key physiological processes and behavioral activities of fish [4]. Temperature changes beyond the optimal range are known to induce various stress related responses in teleosts [5,6,7], such as oxidative stress [8,9], resulting in changes in the fish’s metabolism to regain homeostasis [5].

The fish gut microbiome is associated with a variety of beneficial functions for the host, such as mediating and stimulating optimal gastrointestinal development [10], producing and supplying vitamins to the host [11,12], and improving the host’s nutrient uptake by providing additional enzymatic activities [13]. In particular, herbivorous fish heavily rely on their gut microbiome for degrading cellulose and other complex polysaccharides to enhance the nutritional value of organic carbon sources [14,15]. Aside from optimized nutrient uptake, the gut microbiome is involved in immunomodulation and maintaining mucosal tolerance [10], enhancing the host’s resilience against infectious diseases [13], and the production of anticarcinogenic and anti-inflammatory compounds [16]. Consequently, deciphering gut microbiome dynamics in cultured fish species will play an indispensable role in promoting animal health and aquaculture productivity [17,18].

While the importance of the gut microbiome for the welfare and performance of fish is now widely acknowledged [19,20] and has been extensively studied for potential applications in aquaculture [13,17], there is a substantial lack of information concerning tropical aquaculture species, such as milkfish, and the influence of thermal stress on the gut microbiome and its interactions with the host [21]. With continuous anthropogenic greenhouse gas emissions, predicted shifts in global temperature regimes include a rapid warming of tropical ocean waters [22,23]. Milkfish in outdoor aquaculture are therefore expected to experience warmer conditions in the future. In other species, gut microbiome composition changed considerably in response to temperature. For instance, rearing temperature and the relative abundance of Firmicutes were negatively correlated in rainbow trout [24]. In Atlantic salmon, increasing water temperatures resulted in a shift from generally beneficial lactic acid bacteria to potentially pathogenic *Vibrio* spp. with putative negative implications for host health [25]. Similar findings were published for yellowtail kingfish juveniles, mummichog, and pinfish [26,27]. While the gut microbiome composition has been previously investigated in wild-caught, outdoor pond and open-cage farmed milkfish [28,29,30,31,32], the effect of elevated water temperatures on the gut microbiome and the associated consequences for the fitness and welfare of the host remain unknown.

A variety of physiological biomarkers have been developed for fish and applied as early indicators for stress [33,34]. Apart from biomarkers for the general health and energy status of the fish, such as the hepatosomatic index (HSI) [35] and cellular energy allocation (CEA) [36,37], other parameters are more specific for oxidative stress [9] or aerobic and anaerobic energy production and demands [38,39]. Furthermore, scale cortisol concentrations were shown to reflect the stress history in teleosts, including milkfish [40,41,42]. Although many studies investigated biomarkers in fish with respect to environmental stressors or feeding strategies, only a very limited number of publications attempted to elucidate the interaction between gut microbiome composition and fish physiological biomarkers [43].

The objectives of the present study were to determine the effects of elevated water temperatures within the range of those commonly occurring in outdoor milkfish aquaculture systems [2] on the gut microbiome of juvenile milkfish under controlled experimental conditions, and to assess the relationship between microbiome dynamics and physiological changes. Using 16S rRNA gene amplicon sequencing, we investigated gut microbiome composition and diversity in a thermal exposure experiment, for which milkfish stress and biomarker data were available [40,44], comparing control (26 °C) and elevated temperature (33 °C) treatments over a three week period. We hypothesize that thermal stress will lead to changes in the gut microbiome, which are mediated by host−microbiome interactions and affect the functional role of the microbiome for the host. Our findings provide valuable insights into the so far poorly described milkfish gut microbiome in the context of the host’s physiological state at elevated temperatures, and may contribute to predicting future responses of these animals in outdoor culture systems to climate change.

## 2. Materials and Methods

### 2.1. Experimental Set-Up and Sampling

This study was conducted alongside the thermal stress experiment described in Hanke et al. [40] and Hanke [44]. Briefly, 150 juvenile milkfish obtained from the Brackish Water Aquaculture Development Center in Situbondo (Indonesia) were randomly distributed among six experimental tanks, each integrated into an independent recirculation system (RAS). After an acclimatization period of four weeks at 26 °C, the temperature in three tanks was gradually increased from 26 °C to 33 °C by 1 °C per day over the course of one week and kept constant at 33 °C for three weeks (HT: high temperature treatment), while the remaining three tanks were kept at 26 °C (CT: control treatment). Four to seven fish per tank (for more details see https://doi.pangaea.de/10.1594/PANGAEA.919971) were sacrificed for gut microbiome analysis after the initial temperature increase (d0: day 0), after two weeks (d14: day 14), and at the end of the experiment (d21: day 21). To ensure the presence of as little as possible foreign material in the gut, the fish were not fed 24 h prior to each sampling event. The sacrificed fish were immediately dissected on ice with forceps and scalpels sterilized by ethanol and flaming. The fish’s abdomen was carefully opened, and the internal organs removed. The content of the intestine between the pylori gland and anus was squeezed out and collected in a sterile 2 mL Eppendorf tube, and stored at −80 °C until DNA extraction. To assess the influence of the microorganisms in the tank water or the fish feed on the gut microbiome, 250 mL of tank water were filtered onto 0.2 μm polycarbonate filter (Whatman Nuclepore, Dassel, Germany) at each sampling event and 0.5 g of feed pellets were collected for microbial community analysis and likewise stored at −80 °C.

Additionally, samples of muscle and liver tissue were collected from the same milkfish specimens at each sampling event (d0, d14, d21) for the analysis of the following physiological biomarkers: The activities of key enzymes in metabolic (IDH: isocitrate dehydrogenase, LDH: lactate dehydrogenase, ETS: electron transfer system) and antioxidant related (SOD: superoxide dismutase, CAT: catalase) pathways, the amount of available energy resources (total protein, carbohydrates, lipids) and potential cellular damage due to oxidative stress (LPO: lipid peroxidation) were measured. The hepatosomatic index (HSI) was calculated for each fish as ratio of fish weight and liver weight. Furthermore, regenerated scales that were regrown over the duration of the experiment between d0 and d21 were sampled at d21 for the analysis of scale cortisol to quantify chronic stress levels. These parameters are summarized in the Appendix A
Appendix A and the full data set is available on Pangaea (https://doi.pangaea.de/10.1594/PANGAEA.919971). Further details about milkfish husbandry conditions, sample collection, analytical procedures, and results of the cortisol and biomarker measurements are described elsewhere [40,44]. The experiment was conducted according to German guidelines and regulations regarding animal welfare (permission according to §11 Section 1 clause 1, Tierschutzgesetz; permit number: Az. 522-27-11/0200(132)).

### 2.2. DNA Extraction

DNA extraction followed a phenol-chloroform-isoamyl alcohol protocol modified after Nercessian et al. [45]. Frozen samples were thawed and processed on ice. The gut content was fully suspended in 750 μL CTAB buffer (5% *w/v* cetyltrimethylammonium bromide, 0.8 M NaCl, 120 mM potassium phosphate in DEPC water at pH 8) and transferred to a UV-sterilized bead tube prefilled with 0.3 g of 0.1 mm, 0.25 g of 1 mm and 3 beads of 3 mm diameter glass beads. Filters and feed pellets were directly placed in the bead tube and 750 μL CTAB buffer was added. To the sample in the bead tube, 37.5 μL of 20% SDS (prewarmed to 50 °C) and 75 μL 10% N-Lauroylsarcosin were added and mixed by inverting the tubes. Bead beating was then conducted on the FastPrep-24 Instrument (MP Biomedicals Germany, Eschwege, Germany) at 4 s/m for 20 s. To reduce the foam, tubes were centrifuged for 10 s at 16,000× *g* before adding 750 μL phenol-chloroform-isoamyl alcohol (25:24:1). To ensure a uniform emulsion, tubes were vigorously vortexed twice for 30 s. To separate the aquatic from the organic phase, samples were centrifuged at 4 °C at 16,000× *g* for 10 min. 450 μL of the upper aquatic phase were carefully transferred without disturbing the organic phase to a new 1.5 mL Eppendorf tube filled with 900 μL PEG/NaCl (1.6 M NaCl, 30% *w*/*v* polyethylene glycol 6000 in DEPC water) solution and 2 μL of glycogen solution (5 mg/mL). The tubes were mixed by inverting 10 times to avoid surface layer formation and then stored on ice in the dark for 2 h. The precipitated DNA was pelleted by centrifugation at 17,000× *g* for 90 min at 4 °C. The supernatant was decanted and the pellet was washed with 1 mL ice-cold 70% ethanol by centrifugation at 17,000× *g* at room temperature for 10 min. To fully evaporate the ethanol, pellets were air-dried at room temperature for 10 min. DNA pellets from gut content, fish feed, and water filters were resuspended in 50 μL and in 30µL 1 x TE Buffer (10 mM TRIS-HCl, 0.1 mM EDTA pH 8.0, ThermoFischer Scientific), respectively, on a thermoblock with 400 rpm shaking at 35 °C for 20 min.

### 2.3. 16S rRNA Gene Amplicon Sequencing

DNA extracts were sent to LGC Genomics GmbH (Berlin, Germany) for paired-end amplicon sequencing on the Illumina MiSeq platform (V3 chemistry). Sequencing libraries were prepared using the primer pair 515F-Y (5′-GTGYCAGCMGCCGCGGTAA-3′) and 926R (5′-CCGYCAATTYMTTTRAGTTT-3′) to amplify the V4V5 hypervariable region of the 16S rRNA gene [46]. Primer-clipped paired-end reads were archived using the brokerage service GFBio [47] and are accessible on ENA (PRJEB39140). Further sequence processing steps were conducted together with the sequencing data generated from other samples related to this study on the same sequencing platform with the same library preparation protocol (SRA: SRP155505, ENA: PRJEB33594) using the R package dada2 version 1.8.0 [48]. Briefly, forward and reverse reads were trimmed to 220 bp and 210 bp, respectively, and quality filtered at a maximum expected error rate of 4. Learning of error rates and denoising were conducted using the complete data set for both steps. Reads were then merged and resulting amplicon sequence variants checked for chimeras using default parameters. Only amplicon sequence variants, hereafter referred to as operational taxonomic units (OTUs), between 362 and 399 bp and occurring at least twice in the data set were taxonomically classified using the SILVA reference database version 132 and the SILVAngs web service [49,50,51]. OTUs unclassified on phylum level and those affiliated with eukaryotes, chloroplast and mitochondrial sequences were removed from the analysis. Furthermore, samples with less than 500 sequences were excluded. The final OTU table and associated taxonomy are available on Pangaea (https://doi.pangaea.de/10.1594/PANGAEA.919971).

Amplicon sequencing of the 16S rRNA gene resulted in a total of 3,792,545 quality checked sequences, with on average 40,064 and 29,819 sequences per tank water and fish gut sample, ranging from 13,317 to 100,226 and from 544 to 157,951, respectively, and representing 3567 OTUs in total. No 16S was amplifiable from the fish feed sample.

### 2.4. Statistical Analyses

All statistical analyses and data visualization were implemented in the statistical computing environment R 3.5.1 [52] using the packages vegan [53], nlme [54], and emmeans [55]. R scripts are available on Pangaea (https://doi.pangaea.de/10.1594/PANGAEA.919971)

Alpha diversity of the gut microbiome was assessed based on the inverse Simpson index [56]. No subsampling to equal sequencing depth across samples was performed, since rarefaction curves for this index were saturated at available sequencing depths. The effects of temperature treatment and sampling timepoint were assessed using the log-transformed inverse Simpson index in a general linear mixed model (GLMM) with tank as random factor. Post-hoc pairwise comparisons for significant model terms were implemented using the *emmeans* function.

Beta diversity was assessed based on Bray-Curtis dissimilarities of relative OTU proportions, and visualized using complete linkage hierarchical clustering and principal coordinate analysis (PCoA). The latter was calculated based on square-root transformed dissimilarities, and was further used to calculate community heterogeneity (betadispersion) as distance to centroid in the ordination within groups of samples defined by treatment and timepoint. Differences in betadispersion were tested based on log-transformed distances to centroids using the GLMM and post hoc tests described above.

The effects of treatment and timepoint on gut microbiome composition were tested using permutational multivariate analysis of variance (PERMANOVA) based on square-root transformed Bray-Curtis dissimilarities as implemented in the *adonis2* function. Restricted permutations were used to account for the mixed model design. As observations for each level of the random factor (i.e., tank) and timepoint were not balanced, observations were standardized to six samples either by random selection without replacement if more than six samples were available, or by randomly selecting from the existing samples to fill the missing observations if less than six sample were available. To avoid a bias due to this procedure, the random (re)sampling for the PERMANOVA was repeated 100 times and the average of the R^2^, F, and p-values were reported. As a post hoc test to determine the separation of microbial communities between pairwise combinations of treatment and timepoint, analysis of similarity (ANOSIM) R values were calculated.

The relationship between gut microbiome composition and physiological biomarkers was assessed using redundancy analysis (RDA). Samples with missing biomarker data were excluded. Prior to RDA, rare and low sample coverage OTUs were removed from the data set, i.e., those which did not occur with at least 0.1% sequence proportion in at least three samples. This removal did not alter beta diversity patterns (Mantel test, r = 0.99, *p* < 0.001). Then, the sequence counts were centered log ratio (clr)-transformed, adding a prior of 0.5 to all values. To select the biomarkers best suited to explain patterns in gut microbiome composition, forward model selection was employed considering only one parameter in collinear parameter combinations. In comparison, additional RDAs were performed with individual biomarker parameters as predictors of community composition to assess their total effects.

Furthermore, patterns among individual OTUs depending either on treatment and timepoint or on physiological biomarkers were investigated using clr-transformed sequence counts in a GLMM as described above, followed by pairwise post hoc comparisons, or by calculating correlation coefficients, respectively. OTUs were identified as differentially enriched based on Benjamini−Hochberg corrected p-values for each model term at a significance threshold of 0.05, or as moderately to strongly correlated with biomarker parameters at an absolute Pearson and Spearman correlation coefficient of 0.5.

## 3. Results

### 3.1. Comparison of the Gut Microbiome to Water Microbial Communities

The milkfish gut microbiome was distinct from the microbial communities in the tank water, which formed well separated clusters without any overlap (Figure 1A). Bray-Curtis dissimilarities between gut and water microbial communities per tank at any given sampling event were at least 0.92, and for the majority of samples greater than 0.99. Water communities were dominated by heterotrophic taxa involved in organic matter degradation, such as representatives of the orders Alteromonadales, Flavobacteriales, Oceanospirillales, and Rhodobacterales (Figure 1B). The contribution of dominant gut microbiome OTUs, which cumulatively constitute 70% of the sequences in the fish gut samples, to the tank water at any given sampling event was negligible (<1%, Figure 1B). Likewise, dominant water OTUs were either completely absent or only contributed a minor proportion to the gut microbiome, with the exception of one high temperature (HT) and one control (CT) tank at d14 (Figure 1C). Given the strong separation between gut and water microbial communities, as well as the lack of amplifiable 16S from the fish feed, we classified the observed milkfish gut microbiome as native without external contamination.

### 3.2. Effects of Elevated Temperature on Gut Microbiome Diversity

The milkfish gut microbiome showed a high degree of dominance as indicated by the inverse Simpson index, which ranged between 1.23 and 13.65 with a median of 3.75 (Figure 2A). While no statistically significant differences in the inverse Simpson index were detected between temperature treatments, there was a significant pattern over time, with two to three times lower inverse Simpson indices at the end of the experiment at d21 (Table 1). This pattern was less pronounced for the high temperature treatment, where immediately after the temperature increase at d0 on average lower and less variable inverse Simpson indices were recorded compared to the control treatment.

Gut microbiome composition varied considerably between as well as within temperature treatments and sampling timepoints. PCoA ordination separated microbial communities between sampling events along PCoA1, which captured 33.55% of the variation, while temperature treatments were separated along PCoA2 (Figure 2C). Interestingly, the position of gut microbiome samples in the ordination indicated similar succession patterns over time in both treatments, with a convergence of community composition towards the end of the experiment (d21). Additionally, the size of the ellipses representing the 95% confidence interval of the centroids of each combination of treatment and timepoint suggested a change in within-group community heterogeneity. This was confirmed by betadispersion analysis, which revealed a uniform within-group heterogeneity for the gut microbiome of the control treatment at all sampling events, while in the high temperature treatments, within-group heterogeneity was significantly reduced at d0, but reached values similar to the control treatment again at d14 and d21 (Figure 2B; Table 1).

PERMANOVA provided statistical support for the differences between treatments and succession over time observed in the PCoA ordination, showing strong effects for treatment and timepoint as well as a weaker but significant effect of the interaction of both factors (Table 1). However, only a small fraction of 8.62, 12.84, and 2.94% of the variation in microbiome composition could be explained by treatment, timepoint, and their interaction, respectively. The R values of pairwise ANOSIM tests indicated that microbial communities were only weakly to moderately separated between each combination of treatment and timepoint, with average Bray-Curtis dissimilarities between groups being only marginally higher than those within groups (Table 2). The degree of community separation increased over time in both treatments, while control and high temperature treatments displayed increasing community overlap towards the end of the experiment. In summary, both alpha and beta diversity measures pointed towards strongest differences between control and high temperature treatments immediately after the temperature increase (d0) that diminished over time.

### 3.3. Taxonomic Composition of the Milkfish Gut Microbiome

The milkfish gut microbiome was dominated by the bacterial taxonomic orders Fusobacteriales and Vibrionales, which constituted a high, albeit variable, sequence proportion in all samples (Figure 3A). Fusobacteriales comprised the majority of the sequences in the high temperature treatment at d0 (average sequence proportion 60%) and decreased in their contribution over time at d14 (40%) and d21 (23%). Their proportion in the control treatment was generally lower with 22, 8, and 1% at d0, d14, and d21, respectively. Presumably, Fusobacteriales were gradually replaced by Vibrionales, which displayed the opposite pattern with on average 58, 63, 79% in the control treatment, and 32, 48, and 60% in the high temperature treatment at d0, d14, and d21, respectively. Other taxa, such as Clostridiales and Propionibacteriales, occurred in high sequence proportions of > 20% only in isolated samples compared to a median proportion in the remaining samples of 1% or less. While the primer pair used in this study was theoretically also able to amplify archaeal 16S, only 2 sequences in the whole data set were classified as archaea.

At a higher taxonomic resolution, the gut microbiome was dominated by OTUs affiliated with the genera *Catenococcus*, *Enterovibrio*, and *Vibrio* (Vibrionales), *Cetobacterium* (Fusobacteriales), and *Epulopiscium*, *Romboutsia*, and *Tyzzerella* (Clostridiales). Many of these OTUs were differentially enriched between temperature treatments and sampling timepoints, driving the observed beta diversity patterns (Figure 3B; Appendix A). Overall, 132 OTUs were detected as differentially enriched, of which 77, 123, and 4 OTUs displayed significant differences between treatments, timepoints, and their interaction, respectively. Succession over time was mainly related to differentially enriched OTUs of the genera *Enterovibrio* and *Vibrio*, which exhibited inverse trends. OTUs of the genus *Cetobacterium* showed generally decreasing proportions over time as well as overall increased proportions in the high temperature treatment. Otherwise, differences between treatments were attributed to a taxonomically diverse group of less dominant OTUs, e.g., *Cutibacterium* (Propionibacteriales), *Curvibacter* (Betaproteobacteriales), and one *Vibrio* OTU, which all displayed increased proportions in the control treatment. Furthermore, among the most dominant OTUs, one *Cetobacterium* and two *Enterovibrio* OTUs exhibited different trends over time in the two treatments, with their proportion decreasing sooner in the control than the high temperature treatment. Overall, differentially enriched OTUs constituted the majority of the sequences in most samples (Figure 3C), indicating that succession and temperature effects caused a profound restructuring of the milkfish gut microbiome composition.

### 3.4. Correlation of Gut Microbiome Composition and Physiological Biomarkers

Additional to the effects of the experimental treatments, we explored the relationship between gut microbiome composition and physiological biomarkers. Considering the collinearity patterns among the observed biomarkers, only HSI was selected as the parameter best suited to explain changes in gut microbiome composition in the RDA, with an adjusted R^2^ of 16% (Table 3). While individually also some of the other physiological biomarkers, i.e., superoxide dismutase (SOD), isocitrate dehydrogenase (IDH), lipid peroxidation (LPO), lactate dehydrogenase (LDH), electron transfer system (ETS) and lipid content, were able to explain a low to moderate proportion of the variability in gut microbiome composition, when accounting for collinearity their pure effects were negligible. Other parameters, i.e., carbohydrate and protein content, catalase (CAT), cellular energy allocation (CEA), and scale cortisol concentrations were not related to gut microbiome composition. Correlation analysis between HSI and individual OTU proportions revealed exclusively negative relationships with several *Cetobacterium* OTUs, including a highly dominant one (Figure 3B; Appendix A). Positive correlations were only observed with less dominant OTUs mainly of the genus Vibrio (Appendix A).

## 4. Discussion

This study documented the development of the gut microbiome of juvenile milkfish kept at different temperature regimes under controlled experimental conditions, assessing different external (temperature, feed and water microbial communities) and internal (fish physiology) influences. The pronounced difference between the gut microbiome and the microbial community in the tank water supported the hypothesis of a strong selection for certain microbes by the host. These observations aligned with previous findings, which revealed that fish intestinal microbiota contained only a few free-living environmental bacteria [57] and were derived from host-specific selection pressures within the intestine [11,58]. The taxonomic composition of the microbial community of the tank water was comparable to that of eutrophic coastal marine and pelagic environments [59,60,61]. Dominant OTUs were affiliated with the genera *Marivita* (Rhodobacterales), [*Polaribacter*] *huanghezhanensis*, and the NS3 marine group (Flavobacteriales). Their metabolism characterizes them as aerobic anoxygenic photoheterotrophs and chemoheterotrophs adapted to exploit elevated concentrations of inorganic nutrients and dissolved organic matter [61,62,63,64]. As such, water conditions in the tanks during the experiment [40], including water microbial communities, were representative of the conditions milkfish are likely to encounter in outdoor farming facilities.

### 4.1. Diversity and Composition of the Milkfish Gut Microbiome

The gut microbiome of the investigated milkfish specimens was mainly composed of Proteobacteria (mainly Vibrionales) and Fusobacteria (mainly Fusobacteriales). It is common for the fish gut microbiome to be dominated by only a limited number of bacterial phyla [58]. Previous studies identified the phyla Proteobacteria, Firmicutes, Actinobacteria, Bacteroidetes and Fusobacteria as predominant members of the teleost gut microbiome [13,65]. At a higher taxonomic resolution, *Pelomonas* and *Fusobacterium* spp. were reported as dominant genera in outdoor cultured milkfish [28]. Interestingly, both genera were not detected in the gut microbiome of our experimental milkfish. Instead, we identified dominant OTUs as *Catenococcus*, *Enterovibrio*, and *Vibrio* (Vibrionales), *Cetobacterium* (Fusobacteriales), and *Epulopiscium*, *Romboutsia*, and *Tyzzerella* (Clostridiales). These and other discrepancies to previous observations [28,29,30,31,32] were likely associated with differences in the developmental stage of the investigated fish, fish culturing conditions and methodological biases as with the exception of one study [28], cultivation-dependent methods were used.

Many of the bacterial taxa found in fish guts are reported to facilitate and support digestive processes by providing a variety of enzymes, optimizing nutrient uptake [66,67]. Bacteria from the order Clostridiales are often considered as polymer degraders, fermenting polysaccharides and peptides to yield alcohols and short chain fatty acids (SCFAs) [58]. Studies by Ray et al. [68] and Wu et al. [69] correlated the abundance of *Clostridium* spp. in the gut microbiome of carp with the capacity of cellulose and cellobiose degradation. Although *Clostridium* was not observed in high proportions, other genera from the order Clostridiales contributed considerably to the milkfish gut microbiome with possibly similar biological functions, specifically *Epulopiscium* and *Romboutsia*. These taxa have been identified as prominent members of the gut microbiome in herbivorous and detritivorous fish exhibiting a symbiotic relationship with their host [70,71].

Notably, high proportions of *Cetobacterium* (Fusobacteriales) were found in the gut community, especially in the high temperature treatment after the temperature increase (d0). Fusobacteriales are often associated with the gut microbiome of herbivorous and omnivorous fish, where they are assumed to play a crucial role in carbohydrate degradation [14,15]. They have been described as anaerobic, Gram-negative bacilli, which produce butyrate [72] and vitamins with a potentially vital role in fish welfare [11]. Butyrate is a SCFA, which is an end product of bacterial fermentative carbohydrate degradation and has been attributed beneficial properties for the host, e.g., acting as a potential energy source, promoting growth, enhancing immune response and disease resistance [73]. The most dominant OTU in the data set was closely related to *Cetobacterium somerae*, isolated from the intestine of other fish species (100% sequence identity [74]). Generally, *Cetobacterium somerae* has been found in carnivorous and herbivorous, marine and freshwater fish species, where it is presumed to supplement the host’s vitamin B12 (cobalamin) requirements [12,58,75,76]. Therefore, *Cetobacterium* may be essential for the production of vitamin B12 through fermentative processes in the gut of juvenile milkfish.

Vibrionales dominated the gut microbiome in the majority of the experimental milkfish, in particular those of the control treatment, and were mostly represented by OTUs from the genera *Catenococcus*, *Vibrio* and *Enterovibrio*. *Vibrio* species are often found to be prevalent in and on marine fish and are common members of the gut microbiome of both farmed and wild fish [19]. Here, dominant Vibrionales OTUs were closely related to *Enterovibrio coralii* and various *Vibrio* spp. strains (≥98.7% sequence identity [74]). Although *Enterovibrio coralii* and *Vibrio* spp. have been identified as potential fish pathogens [77], we deem it unlikely that the strains present in the milkfish gut were pathogenic. Considering their high sequence proportion and the overall healthy state of milkfish throughout the experiment (no starvation or exposure to oxygen stress), we assume that the observed strains were likely harmless commensal or even mutualistic bacteria, without negative implications for the animal’s welfare [19,78]. Indeed, isolation-based studies have identified *Vibrio* strains as common members of the milkfish microbiome during late larval and juvenile stages, with potential probiotic properties [29,30,31]. A great variety of digestive enzymes have been described for the genus *Vibrio*, such as amylase, protease, lipase [79] and chitinase [80], all of which may enhance the host’s digestive capabilities and improve nutrient uptake [19,81], thereby making *Vibrio* an important member of the functional gut microbiome of milkfish.

### 4.2. Temperature Effects on the Milkfish Gut Microbiome

We observed pronounced changes in the composition and diversity of the milkfish gut microbiome in both control and high temperature treatments over time. In the control treatment, the drastic increase in the proportion of *Vibrio* OTUs, resulting in the observed decrease in alpha diversity, was the most prominent temporal change. Numerous studies document how teleost gut microbiota are altered by diet, bacteria in the water column, and environmental factors such as salinity, pH or temperature [13,19,20,75]. As the fish in the experiment were kept under stable conditions with respect to water temperature, oxygen levels, salinity and feed [40], these parameters are unlikely to explain the observed changes in the gut microbiome of the control treatment. Instead, such changes might be attributed to the natural succession of the gut microbiome during the ontogenetic development of the fish and associated changes in dietary requirements and metabolism, and may have been shaped by the host’s internal selection pressures [11,82,83]. Additionally, temporal changes in the milkfish gut microbiome may be related to the decrease in stocking density, as fish were removed at each sampling event [84,85].

Before the first sampling for microbiological analyses (d0), the fish of the high temperature treatment experienced a gradual temperature increase. This temperature increase resulted in an elevated stress response, but also increased growth [40,44], as well as a pronounced restructuring of the gut microbiome: communities were more uniform, with high proportions of *Cetobacterium* and *Enterovibrio*, and a reduced alpha diversity. We postulate that the highly congruent changes in gut microbiome diversity and community heterogeneity at d0 were mediated by host−microbiome interactions in response to the observed effects of the temperature increase on milkfish physiology [40,44]. Indeed, we detected a strong correlation between HSI (hepatosomatic index), as proxy for the energy status of the milkfish host, and gut microbiome composition. In this relationship, HSI is representative for a suite of highly correlated biomarkers (LPO: lipid peroxidation, SOD: superoxide dismutase, IDH: isocitrate dehydrogenase) that are further linked to oxidative stress [9]. Specifically, the high proportion of *Cetobacterium* OTUs at d0 in the high temperature treatment coincided with lower HSI, but higher SOD, LPO and IDH measurements [44]. Considering the metabolic capabilities of the close relative *Cetobacterium somerae*, it is possible that its ability to produce SCFA and vitamin B12 additional to its role in aiding digestion [12] may alleviate the stress response of the milkfish to the temperature increase. For instance, SCFA and vitamin B12 have been implied to counteract oxidative stress by inhibiting inflammatory responses and acting as antioxidant, respectively [16,86].

After the initial temperature increase, the subsequent changes in the diversity and composition of the gut microbiome at constant high temperature suggest a parallel development and final convergence with the control treatment, reflected in the increasing dominance of *Vibrio* OTUs. A similar trend was observed for the milkfish host, where whole-body performance (growth) and physiological biomarkers [40,44] indicated that the milkfish might have started to habituate to the elevated temperature conditions. This conclusion is therefore supported by our observations on the gut microbiome. Furthermore, it seems to explain the lack of correlation between gut microbiome composition at d21 and the regenerated scale cortisol concentration, as this parameter provides a retrospective view of a specific time period and therefore represented the cortisol produced throughout the whole period between d0 and d21 [40].

Notably, physiological biomarkers, which were not strongly correlated with gut microbiome composition (LDH, ETS, CAT, CEA, and total lipids, carbohydrates, and protein), also exhibited less pronounced temperature treatment and/or timepoint effects [44]. It is therefore possible that the previously described correlations between gut microbiome and host physiology may be confounded by congruent but independent patterns between treatments and over time. For instance, as milkfish are poikilothermic ectotherms, their body temperature depends on the ambient conditions. As temperature is furthermore known to be one of the most important drivers of microbial community composition [87], changes in the gut microbiome may also be directly related to the temperature change in the high temperature treatment. However, given the strong documented interactions between host and gut microbiome [20] it appears unlikely that temperature alone would determine gut microbiome composition. Yet, temperature may modulate physiological effects on the gut microbiome [88]. Similarly, additional environmental factors would need to be considered in a coastal aquaculture setting, where the majority of the milkfish production is taking place in open cages or pens [3].

## 5. Conclusions

We demonstrated that even gradual temperature changes within a range commonly occurring in outdoor milkfish aquaculture systems lead to a major restructuring of the milkfish gut microbiome. This trend seemed to be mediated by host−microbiome interactions in accordance with a pronounced physiological response of the milkfish to the temperature increase. Given the predicted functional role of the affected bacterial taxa for the host, the gut microbiome may be involved in alleviating thermal stress, presumably contributing to the habituation of the milkfish to the elevated temperature over time. Our findings emphasize the importance of the gut microbiome and host−microbiome interactions for fish welfare and aquaculture management, especially in future ocean scenarios in a warmer world.

## Figures and Tables

**Figure 1 microorganisms-09-00005-f001:**
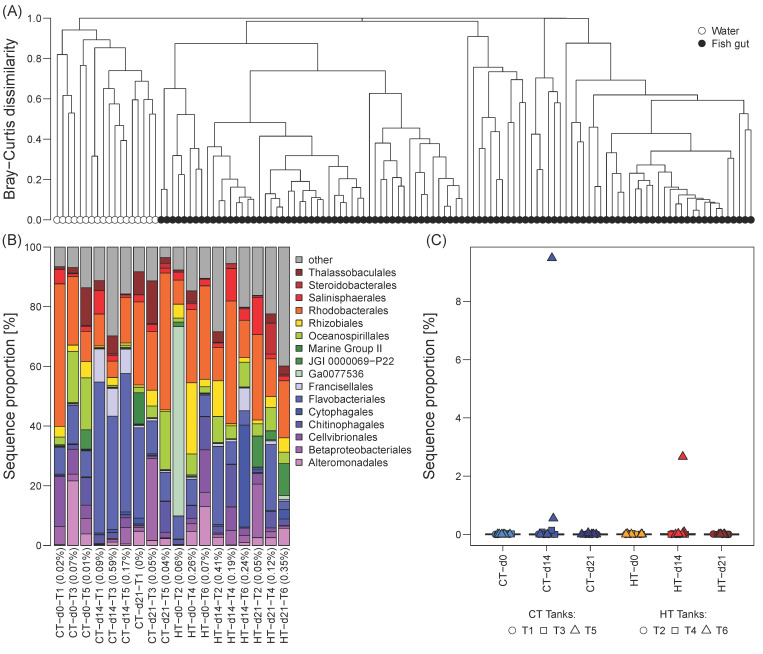
Comparison of microbial community composition of the tank water and the milkfish gut. (**A**) Complete linkage hierarchical cluster diagram based on Bray-Curtis dissimilarities of relative OTU proportions, ranging between 0 (most similar) to 1 (most dissimilar). (**B**) Taxonomic composition of water microbial communities at order level. Percentages in parenthesis indicate contribution of OTUs that cumulatively comprise 70% of the gut microbiome in any of the sampled fish per tank and sampling timepoint to the water community. (**C**) Contribution of the OTUs that cumulatively comprise 70% of the water community to the fish gut microbiome in each tank and at each sampling timepoint. Temperature treatments: control temperature at 26 °C (CT) and high temperature at 33 °C (HT). Sampling timepoints: 0, 14, and 21 days after the temperature increase.

**Figure 2 microorganisms-09-00005-f002:**
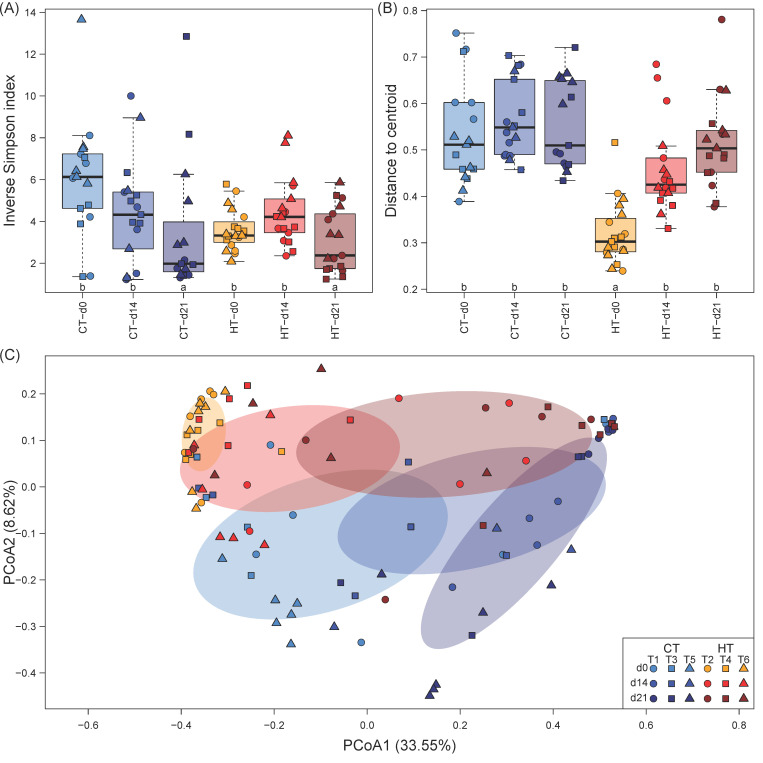
Alpha and beta diversity of the milkfish gut microbiome at different temperature treatments and sampling timepoints. (**A**) Alpha diversity (inverse Simpson index). (**B**) Betadispersion (distance to centroid). In A and B, lower case letters indicate significant differences between groups of samples. (**C**) Principal coordinate analysis (PCoA). Percentages in parenthesis indicate the amount of variation shown on each axis. The shaded ellipses show the 95% confidence interval of the centroid of each group. Temperature treatments: control temperature at 26 °C (CT) and high temperature at 33 °C (HT). Sampling timepoints: 0, 14, and 21 days after the temperature increase. Each treatment consisted of three independent tanks (T: tank number).

**Figure 3 microorganisms-09-00005-f003:**
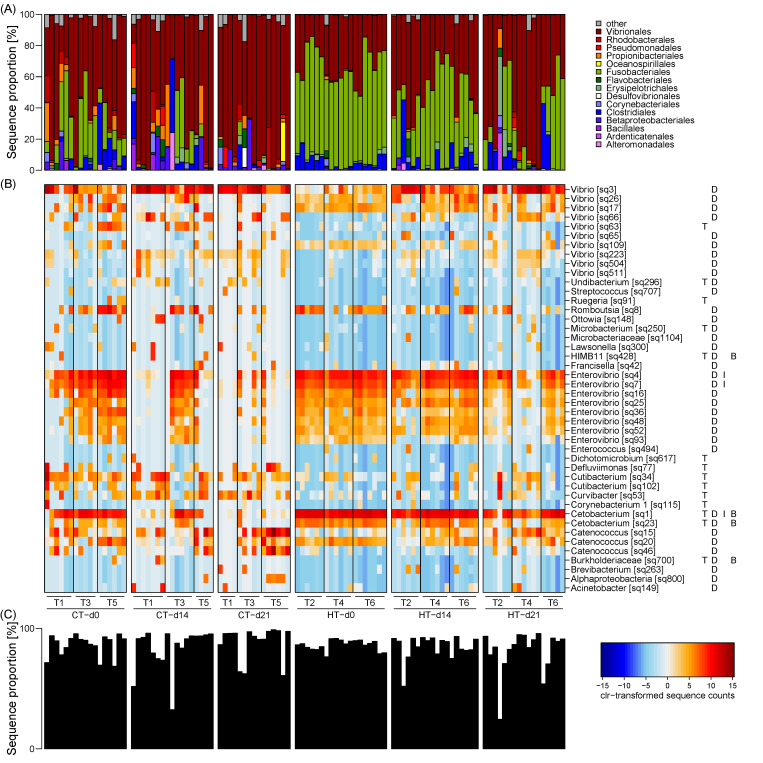
Taxonomic composition of the milkfish gut microbiome and differentially enriched OTUs at different temperature treatments and sampling timepoints. (**A**) Taxonomic composition at order level. (**B**) Heatmap of clr-transformed sequence counts of differentially enriched OTUs. Only dominant OTUs with at least 1% sequence proportion in one sample are shown. Genus level taxonomy and OTU names are provided on the right, followed by capital letters indicating whether differences were observed between treatments (T), timepoints (days; D), the interaction of these two factors (I), or whether they were correlated to the hepatosomatic index (biomarker; B). (**C**) Total contribution of differentially enriched OTUs to the microbiome composition. Temperature treatments: control temperature at 26 °C (CT) and high temperature at 33 °C (HT). Sampling timepoints: 0, 14, and 21 days after the temperature increase. Each treatment consisted of three independent tanks (T: tank number).

**Table 1 microorganisms-09-00005-t001:** GLMM and PERMANOVA results to assess the effects of temperature treatment and sampling timepoint on the alpha diversity (inverse Simpson index), betadispersion (distance to centroid), and composition of the milkfish gut microbiome.

Parameter	Model Terms	numDF ^1^	denDF ^2^	R^2^ [%]	F-Value	*p*-Value
Inverse Simpson index (GLMM)						
	Treatment	1	4		0.462	0.534
	Timepoint	2	93		7.328	0.001
	Treatment:Timepoint	2	93		2.926	0.059
Distance to centroid (GLMM)						
	Treatment	1	4		57.134	0.002
	Timepoint	2	93		22.335	<0.001
	Treatment:Timepoint	2	93		12.533	<0.001
Community composition (PERMANOVA)						
	Treatment	1	102	8.62	11.633	0.001
	Timepoint	2	102	12.84	8.668	0.001
	Treatment:Timepoint	2	102	2.94	1.987	0.009

^1^ Numerator degrees of freedom; ^2^ denominator degrees of freedom.

**Table 2 microorganisms-09-00005-t002:** Separation of gut microbial communities between treatments and sampling events. Average Bray-Curtis dissimilarity between (upper triangle) and within (diagonal) groups and ANOSIM R values (lower triangle, in parentheses).

	CT-d0	CT-d14	CT-d21	HT-d0	HT-d14	HT-d21
**CT-d0**	0.63	0.77	0.86	0.57	0.59	0.72
**CT-d14**	(0.30)	0.69	0.71	0.82	0.74	0.66
**CT-d21**	(0.51)	(0.06)	0.68	0.94	0.85	0.70
**HT-d0**	(0.41)	(0.69)	(0.89)	0.23	0.40	0.68
**HT-d14**	(0.15)	(0.39)	(0.65)	(0.20)	0.46	0.62
**HT-d21**	(0.26)	(0.04)	(0.17)	(0.56)	(0.24)	0.59

**Table 3 microorganisms-09-00005-t003:** RDA using individual biomarkers as predictors for gut microbiome composition.

Parameter ^1^	R^2^ [%]	Adjusted R^2^ [%]	numDF ^2^	denDF ^3^	F-Value	*p*-Value
HSI	17.25	16.38	1	96	20.005	0.001
SOD	9.24	8.29	1	96	9.771	0.001
IDH	6.70	5.73	1	96	6.891	0.002
LPO	4.46	3.46	1	96	4.478	0.004
LDH	3.50	2.49	1	96	3.477	0.010
ETS	2.94	1.93	1	96	2.908	0.026
Lipids	2.73	1.71	1	96	2.691	0.027
Carbohydrates	1.17	0.14	1	96	1.136	0.272
Protein	1.01	< 0	1	96	0.984	0.332
CAT	0.76	< 0	1	96	0.734	0.543
CEA	0.46	< 0	1	96	0.442	0.899
Scale cortisol	3.12	< 0	1	30	0.966	0.398

^1^ See footnote to Appendix A; ^2^ numerator degrees of freedom; ^3^ denominator degrees of freedom.

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
