# Peer review of "Effects of Thermal Stress on the Gut Microbiome of Juvenile Milkfish (Chanos chanos)"

_microorganisms, 2020, doi:10.3390/microorganisms9010005_

Round 1
Reviewer 1 Report
The paper “Effects of thermal stress on the gut microbiome of 2 juvenile milkfish (Chanos chanos)” by Hassenruck et al. reported on the effect of an experimental thermal stress (3 weeks at 33°C) on the gut microbiota of Milkfish. The resulted microbiota modification was correlated with biomarkers related to stress.
The paper is very clear and well written, the experiments are well described, and the result are well supported by experimental data, the statistical tools are appropriate and the data are well discussed. The conclusions are of interest, microbiota modifications in controlled conditions are very important data as starting points for interpretation of real events.
For these reasons, in my opinion this paper is suitable to be published in Microrganisms.
I have only minor suggestions for the authors:
Line 104-105 please detail the number of fish collected for each tank
Line 219-220 move to Materials and methods section.
Line 240-282 In the PERMANOVA the effect of “Tank” on the microbiome composition might be evaluated.
Author Response
Thank you very much for the positive assessment of our manuscript. Below are our responses to your detailed comments.
Line 104-105 please detail the number of fish collected for each tank
ANSWER: Due to failed DNA extractions and library preparation, the number of fish gut microbiome samples that could be analyzed varied between tanks. As the exact number of fish per tank can be accessed in the data archived on Pangaea we would like to refrain from detailing this information in the main text. Instead we included the Pangaea link.
Line 219-220 move to Materials and methods section.
ANSWER: We moved these sentences to the end of the amplicon sequencing paragraph in the methods section.
Line 240-282 In the PERMANOVA the effect of “Tank” on the microbiome composition might be evaluated.
ANSWER: We accounted for the effect of tank by using a mixed model approach. As we were primarily interested in the temperature effect over time, we included tank as random factor to account for the variability among the independent RAS tanks. As such, the effect of tank is internally evaluated and accounted for by our PERMANOVA model.
Reviewer 2 Report
The paper well describe the topic of the research and will explain the results.
However, I suggest to include the supplementary figures in the main text.
Minor modifications
Lane 154: the title 2.3.16. S amplicon sequencing should be 2.3. 16S rDNA amplicon sequencing otherwise it is not understandable at all
lane 220: what do you mean by “40 064 and 29 819 sequences per water and fish gut sample,”
lane 227: Bray-Curtis dissimilarities: what does the number indicate? The dissimilarity is high or low?
Author Response
Thank you very much for the positive assessment of our manuscript. Below are our responses to your detailed comments.
However, I suggest to include the supplementary figures in the main text.
ANSWER: We included the SI figure 1 in the main text as requested.
Lane 154: the title 2.3.16. S amplicon sequencing should be 2.3. 16S rDNA amplicon sequencing otherwise it is not understandable at all
ANSWER: This typo must have been introduced during the formatting of the manuscript after upload. We now changed the subsection title to “16S rRNA gene amplicon sequencing”.
lane 220: what do you mean by “40 064 and 29 819 sequences per water and fish gut sample,”
ANSWER: Upon request of the other reviewer, this sentence was moved to the methods section. With these numbers we are referring to the average number of sequences generated per tank water sample and fish gut sample, respectively. We then provide the range in sequencing depth in the second part of the sentence.
lane 227: Bray-Curtis dissimilarities: what does the number indicate? The dissimilarity is high or low?
ANSWER: Bray-Curtis dissimilarities range between 0 (most similar) to 1 (most dissimilar). We added this explanation in the caption of the corresponding figure.